# Bitcoin: A life in crises

**Jevgeni Tarassov[1], Nicolas Houlié [2]\***

**1** Independent researcher, Zug, Switzerland, **2** ETH Zurich, Institute of Geophysics, Seismology and Geodynamics, Zurich, Switzerland

\* nhoulie@alumni.ethz.ch

## Abstract

In this study, we investigate the BTC price time-series (17 August 2010–27 June 2021) and show that the 2017 pricing episode is not unique. We describe at least ten new events, which occurred since 2010–2011 and span more than five orders of price magnitudes ($US 1 –$US 60k). We find that those events have a similar duration of approx. 50–100 days. Although we are not able to predict times of a price peak, we however succeed to approximate the BTC price evolution using a function that is similar to a Fibonacci sequence. Finally, we complete a comparison with other types of financial instruments (equities, currencies, gold) which suggests that BTC may be classified as an illiquid asset.

**Data Availability Statement:** The data used in this study can be found at the following URL: https://finance.yahoo.com/quote/BTC-USD/.

**Funding:** The author(s) received no specific funding for this work.

## 1. Introduction

As of today, the combined market value of the 5 most popular cryptocurrencies (CCs) is > $1500 bn (>$950 bn for BTC alone); a number similar to the market cap of Amazon, and larger than those of Tesla or Facebook. In the light of its high historical volatility (90d historical volatility ~ 100%), irregular market trading volumes, and because its main underlying is yet unknown, classifying BTC (and some of the other CCs) for risk assessment remains necessary. Although it was intensely scrutinized, it is yet unclear whether BTC should be treated as a commodity (volatile and liquid), a currency (stable and liquid), an equity (variably liquid and variably volatile), or whether it should be receive a singular definition for each investment context [1–7]. Our main objective is to contribute to this debate by focusing on the study of time-series.

With the increasing speed and improving reliability of financial apps-based services, and with a strong editorial presence in economic arenas, crypto currencies are poised to gain an ever-greater base of users and services [8, 9]. In the wider context of blockchain development, CC may help in trading goods and services across political boundaries, avoiding fees imposed by financial intermediaries, and in hedging uncertainties on financial markets [10]. Even though some progress has been made through the understanding of the bitcoin (BTC) price structure [2, 11], the business model remains largely opaque. The overall opacity surrounding CC trades quickly triggered informal criticisms and, soon after, many warnings issued by a wide range of actors, from intelligence services [12] to market regulators [13].

In November 2017, the price of BTC has unexpectedly risen for a month by an average 1.8% daily, leading to the biggest and most widely publicized exponential price rise in

**Competing interests:** The authors have declared that no competing interests exist.

cryptocurrencies. Despite the fact that holding BTC carries some volatility risk (implied volatility > 100%; [14], large cohorts of market participants seek to invest in BTC and crypto-currencies. This has left banks and other financial services firms scrambling to deploy financial services (custodianship, cold storage, BTC payment services) and investment products (like funds, derivatives, and complex structured notes) into the market to capitalize on this trend, which in turn exposed them to non-trivial challenges, both regulatory and economical. Most difficulties were due to the nature of the BTC price process, which did not lend itself to straightforward Black-Scholes-Merton modelling. This further encourages us to try and understand 1) the price dynamics of BTC and 2) under which assumption(s) the risk of hold-ing BTC exposure should be modelled.

In this study, we use numerical methods such as time-series analysis, which proved their efficiency in many other scientific fields such as finance forensics or geophysics. Time-series analysis allows the uncovering of intrinsic parameters (and their dynamics) of a time evolving phenomenon, and we treat the BTC price time-series as we would any seismological or meteo-rological records, further comparing them against a model we deem appropriate. In order to get the most of this approach, we carried out our analysis is in both time and frequency domains. In the time domain, trends, amplitudes and scattering can be quantified, while in the frequency domain, hidden discontinuities and periodicities can be explored. By combining both approaches, our aim has been to detect market price discontinuities, irregularities of prices, large changes of market capital (cash influx), large buy/sales and crowd effects in the context of a market that is not immune to other financial information available to the greater public. Finally, we fit BTC prices with a so-called *Hockey Stick Function* (HSF) and suggest that one, same, recurring dynamic fuels all BTC price surges. We hope our findings will help to characterize the nature of BTC for risk mitigation purposes.

## 2. Data

In this study, we use daily ('Open') price data freely available on the Yahoo Finance® website, in order to determine long-term dynamics of the BTC market value. The dataset used in this study starts on 17 Aug 2010 and ends on 27 June 2021. We compared these prices with other BTC price feeds, such as those provided by Bloomberg® and Market Map® (Morningstar FOREX prices), and found some differences at given times as observed before [15]. For instance, inter-exchange Bitcoin price differences did not exceed 500 USD during fall 2017-winter 2018 when the price passed USD 10'000 for the first time (Fig 1). As we focus on large changes of prices over long time spans (>2 days), we have made sure that using the other sources for prices would not have led us to different conclusions.

## 3. Results

### 3.1. First level analysis

As the analysis in the frequency domain does not require pre-processing steps (detrending, etc.), we first computed periodograms for both the complete and some subsets of the dataset. For this, we used openly distributed *R* packages [16–18]. The frequency domain approach allows the detection of discontinuities and periodic signals for a given time-series (for this reason, fre-quency analysis may be used to detect e.g. fraud such as price manipulations). Here, we use this tool in order to detect price peaks which may be hidden in either high-amplitude noise or low amplitudes. For example, in Fig 2B, we identify the events 1, 2 and 3 highlighting the correspon-dence between time and frequency analysis. Event 2 corresponds to the period during which BTC passed US$150 (those events are also highlighted in Fig 1). Fig 2 shows that there is no peri-odicity of the BTC prices (i.e. we cannot see continuous horizontal lines, please refer to S1–S9

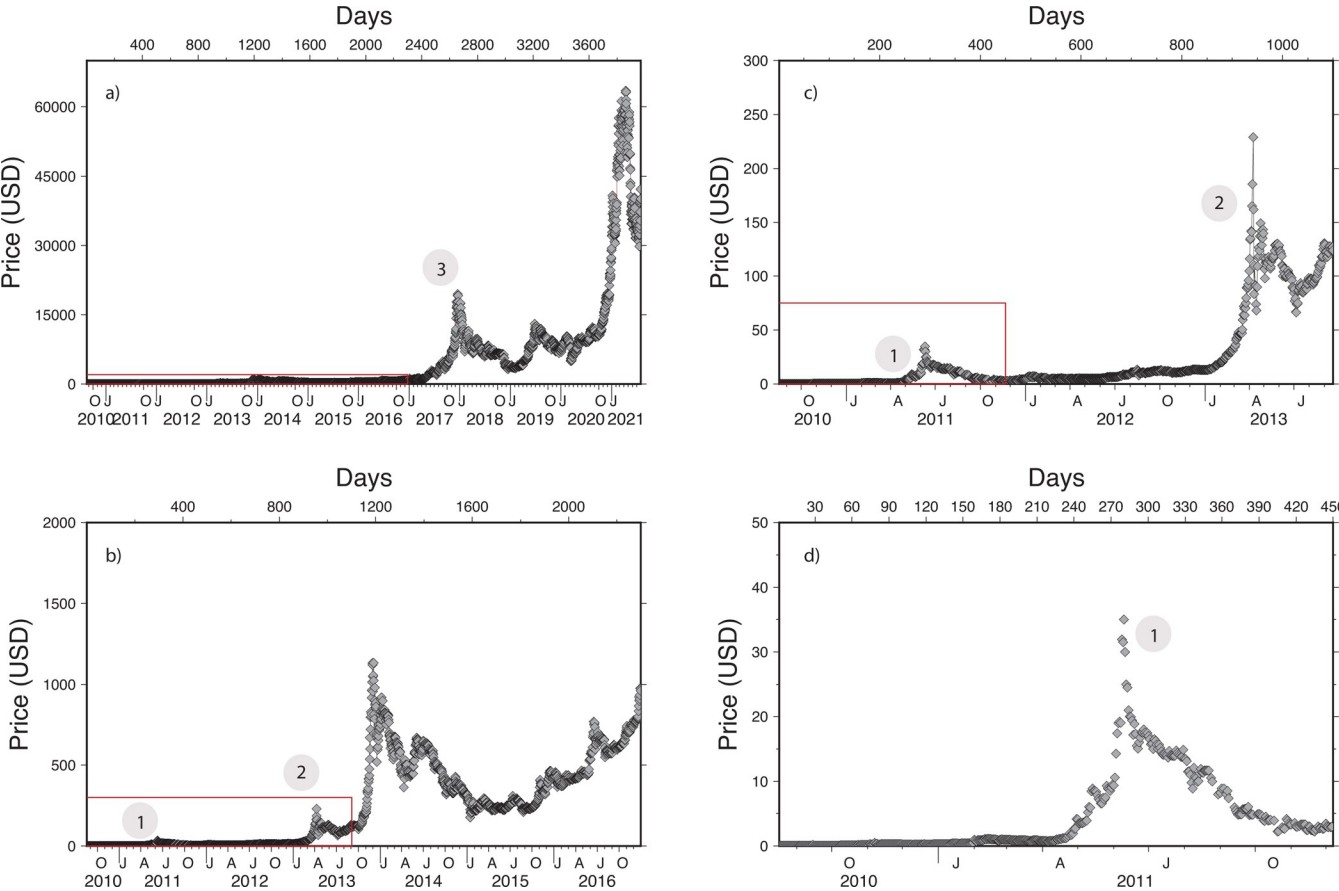

**Fig 1.** Price of BTC between 2010-08-17 and 2021-07-31 (a) for various periods (b-d). Many periods show the episodes of price increase (and decrease) of similar shape. For reference the same events are highlighted in Fig 2 using same labels.

Figs for a purely periodic time-series), indicating that the determination of the asset price does not include a cyclical component. The discontinuities visible in Fig 2A can be linked to price peaks of various amplitudes, therefore suggesting that the price peak of fall 2017-winter 2018 is not unique. Finally, Fig 2, by zooming in on specific periods, also suggests the peak prices to be in fact composed of collections of BTC prices discontinuities (green vertical lines or red areas in Fig 2). All those observations suggest the BTC may suddenly become less liquid, although the demand remains high, leading to a price increase by gain of interest from the investors. As those price changes are mostly positive, we can exclude the hypothesis that price variations were due to the discovery of large numbers of BTC (through mining). In order to provide the reader with a point of reference regarding this technique, we computed periodograms (S1–S9 Figs) for synthetic time-series and popular stocks (ABB, Tesla-TSLA, Gamestop-GME).

Secondly, we focus on the statistical characteristic of the price time-series. We first test whether BTC prices follow the Bendford's law using the "*BenfordTests*" R library [19]. This method is commonly used for forensics analysis of price structure and income tax data analysis [20] and in the case of BTC it may highlight variations of liquidity. As BTC prices range 4 orders of magnitudes, they qualify for this kind of analysis. Chi$^2$ test analysis shows that the prices of BTC do not comply with Benford's law ($\chi^2 = 357$ for n = 8) as the occurrence of "*6*"s for the first digit is occurring too many times (approx. +50% excess; Fig 3). Further analyses would be necessary in order to identify the cause(s) of this observation.

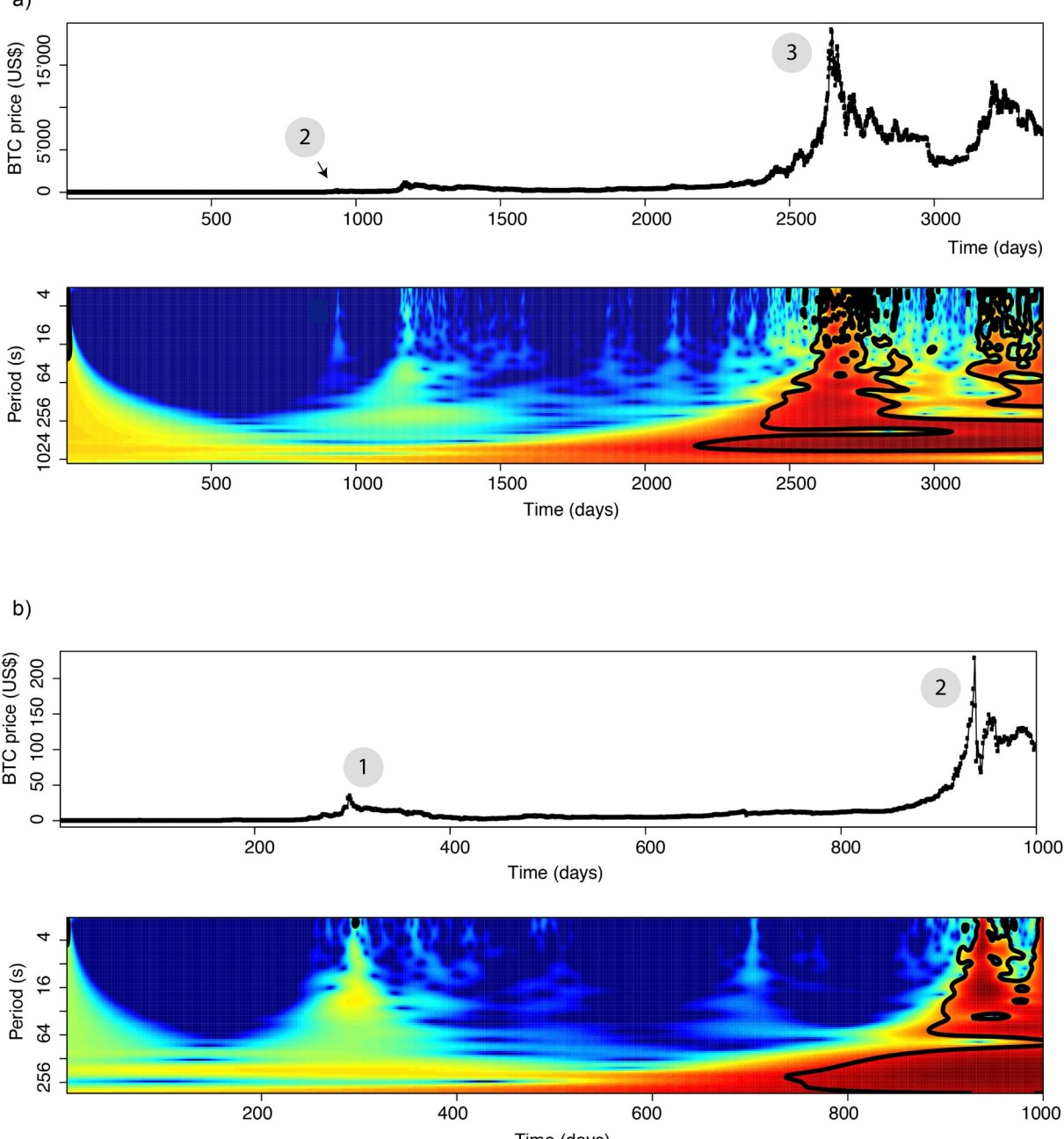

**Fig 2. Frequency analysis of the BTC price history.** a) Whole BTC price time series and associated periodogram for the whole dataset and b) and for the first 1000 days of the dataset (i.e. 2010 until 2014). High amplitudes levels (yellow to red) highlight discontinuities of price, change of frequency contents. One can note the high number of small event (green to yellow within blue areas) occurring in many occasions since 2012. Two classes of events can be distinguished: those which involve long-periods signals like it was seen for earthquake propagation [53], and those which are only small scale discontinuities (e.g. at approx. 700 days on panel b).

## 3.2. The fall 2017 price peak

During the fall 2017 the BTC price tripled, its 30-day volatility was up to 180% (Fig 1), and daily returns sometimes exceeded 10%. Such price changes were faster than base-e

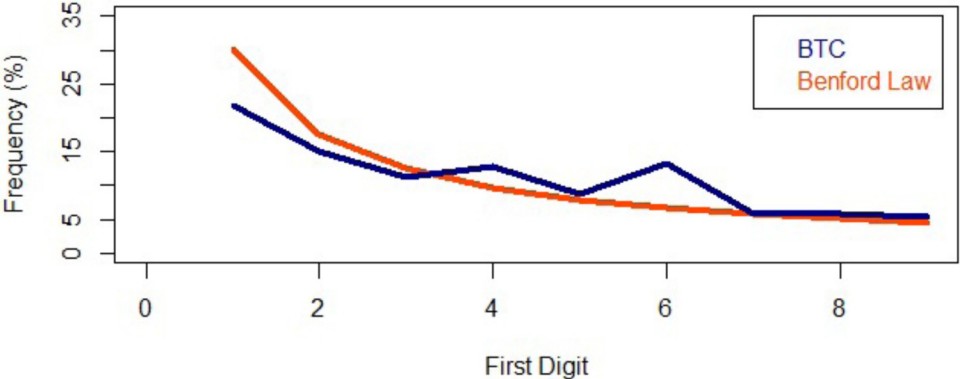

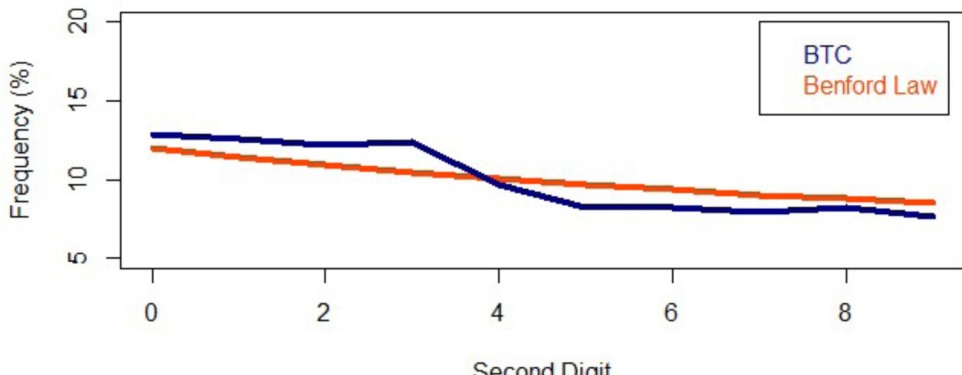

**Fig 3. Comparison between Benford law distribution (orange) and BTC prices (blue) since 17.08.2010.**

exponential, soon looking like a Hockey Stick Function (HSF). HSF is a function parented to the Pascal triangle [21] and Fibonacci series; it can be understood as an exponential function of increasing base. Interestingly, and despite the many opportunities, this function is seldom used to describe natural phenomena. Concentration of $CO_2$ in the Earth atmosphere [22], the global temperatures [23] but also financial transfers in the context of migrations of populations [24] or pandemic developments [25] are good candidates to be modelled by such a function. As HSF seems to be appropriate to variables resulting from crowds increasing in size, we choose to test it on the BTC time-series. We start by modelling the price peak of the fall 2017 as follows:

$$P_i = \left( \sum_{k=i-2}^{i-1} P_k \right) \ (i > 1) \qquad\qquad \text{Eq (1)}$$

Where $i^{th}$ price is determined from the two previous (*i-1* and *i-2*). This formulation is flexible because it can be tuned to fit amplitudes of the signal independently of the time interval used. Eq (1) is of a form similar to Fibonacci series, and like them it is bounded neither in time nor in amplitude. Therefore, the use of this function does not enable us to predict the maximum price of a BTC, nor the time of such a peak. Finally, we must select the periods of interest within the complete dataset to select the peak price of each data subset. In order to calibrate an HSF profile to our dataset and include information on time and

amplitude, we have had to find two input parameters ($P_1$ and $P_2$) which describe the evolution of the prices, as well as a sampling interval ($dT$) which helps define the speed of price increase. To reach a robust solution, we calibrate data against Eq (1), using a brute-force scaling approach, to match both the price amplitudes and the duration of development of each event. For instance, one can satisfactorily approximate the pricing episode of 2017 with $P_1$ = 500 and $P_2$ = 503 (this initial value for $P_1$ fits well with the average value of BTC for the year 2016: approx. \$550 +/- 150 USD) when using daily prices. We show the results of our analysis in Fig 5.

### 3.3. Data fit of secondary episodes (2010–2021)

While it is well known that BTC has been volatile during the fall of 2017, one rarely considers that BTC had already experienced similar pricing episodes composed of 1) a sharp rise of the price (~50–70 days), followed by 2) a short stagnation and 3) a readjustment over several weeks (>100 days). Such a price increase may seem similar to the development of a Minsky-Kindleberger bubble [26–28] except the final price level is not reduced to the pre-surge price. Three of those events have been documented so far: one in 2012 and two in 2013 [29, 30]. Using the approach described above, we found at least 10 additional events which occurred between 2010 and 2021. We have compared those events together by normalizing both their amplitudes and time frames. We show here that regardless of the maximum price, the relationship between duration and price change is strongly consistent, suggesting some self-similarity in the BTC time-series (Fig 4). This observation allows us to apply our approach to all new events found so far.

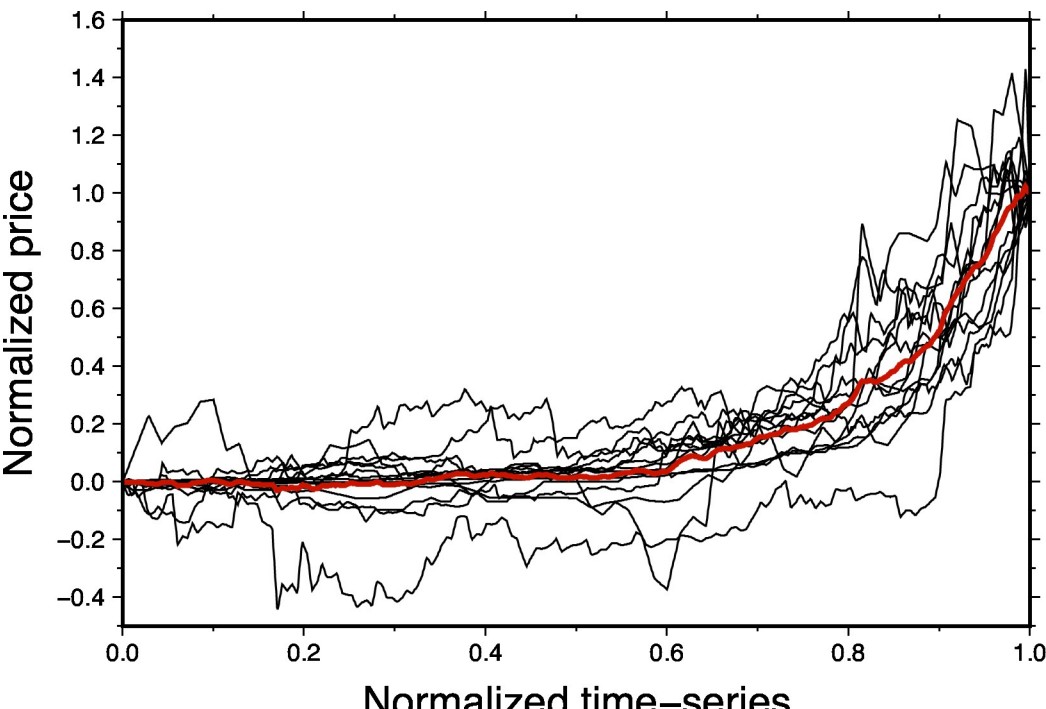

**Fig 4. Time- and Price- normalized data segment preceding price peaks.** Average price history is plotted using a red line. From a price of less than USD1 to > USD 55k, events shows self-similarity. All prices increases are contained within a timeframe of 50–70 days (normalized time ~ 0.7).

We then approximate each event using a HSF by determining the values of $P_1$, $P_2$ and sampling rates. We display the results of all data fit in Fig 5. The quality of each data fit was assessed using their Root Mean Square (RMS) value; distance between each price ($x$) at the time $i$ and the best-fitted model price ($m$) histories:

$$RMS\,(USD) = \sqrt{\frac{1}{N}\sum_{i=1}^{N}\left(x_i - m_i\right)^2}$$

Eq (2)

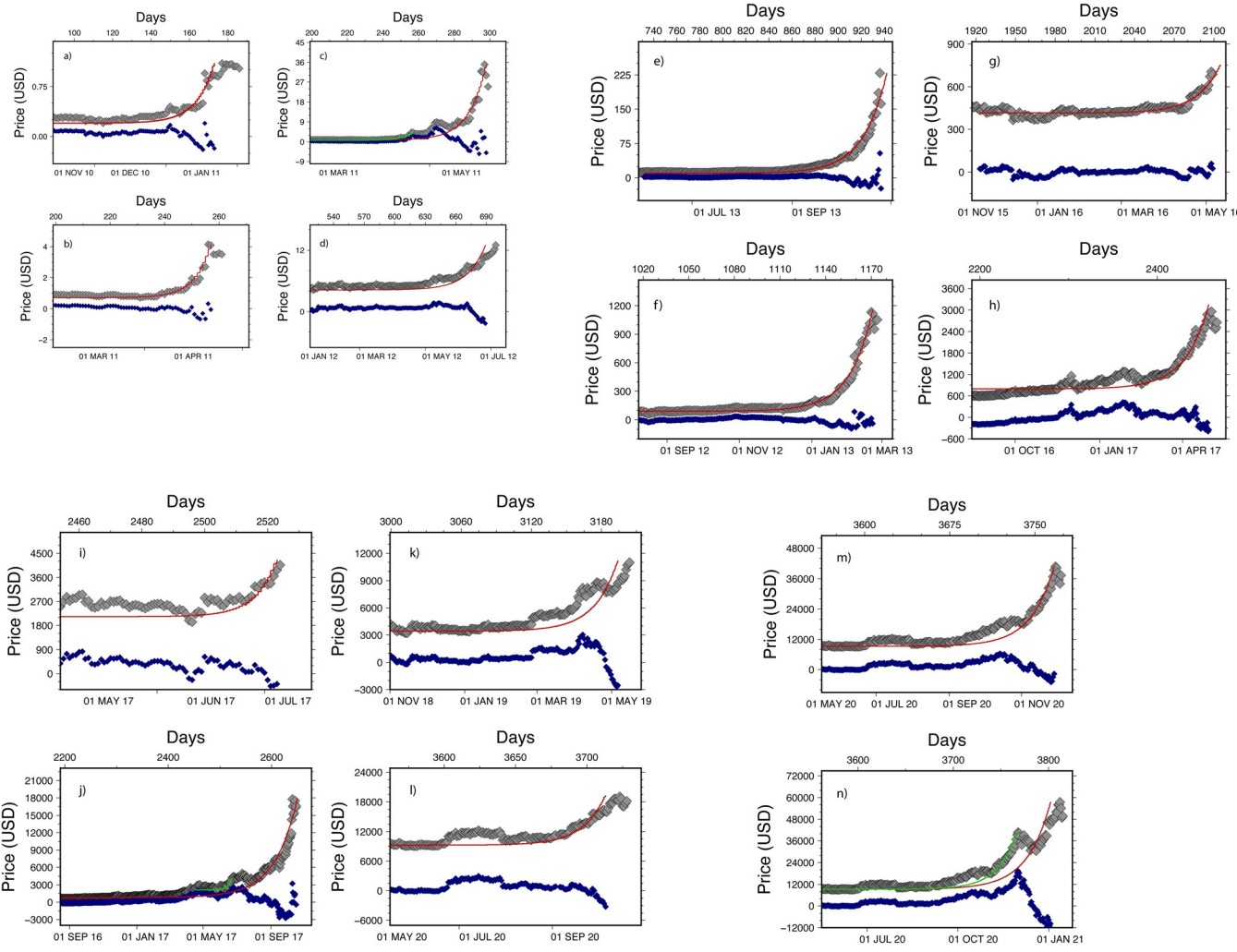

**Fig 5. BTC pricing events presented in this study.** Observed prices are plotted using grey diamonds triangles, modelled prices using red lines. Preceding events are shown using green lines. Residuals are shown using blue diamonds. All curves and residuals parameters are shown in Table 1. BTC pricing events presented in this study. Observed prices are plotted using grey diamonds triangles, modelled prices using red lines. Residuals are shown using blue diamonds. All curves and residuals parameters are shown in Table 1. BTC pricing events presented in this study. Observed prices are plotted using grey diamonds triangles, modelled prices using red lines. Preceding events are shown using green lines. Residuals are shown using blue diamonds. All curves and residuals parameters are shown in Table 1. On panel j), we show for comparison the fitted curve (in green) shown in panel i). BTC pricing events presented in this study. Observed prices are plotted using grey diamonds triangles, modelled prices using red lines. Preceding events are shown using green lines. Residuals are shown using blue diamonds. All curves and residuals parameters are shown in Table 1. Panel m) shows that more research needs to be done to understand the mechanism of each event. If the data of panel m) can be well modeled, the next peak is more difficult to fit (red line; Table 1). Here we have unique opportunity to see to simultaneous events and may need to be studied in more detail.

Table 1 shows the price changes and *RMS* for each event shown in Fig 5. In most cases, we can successfully explain approx. 75% of the data signal (Table 1) or more. The price change over each period is > 77% (median ~ +80%) with a minimal value of 48% for event 4. After each peak, BTC deprecated, but its value was never lower than the values preceding the peak. We can compare this remarkable behaviour to data histories (time-dynamics) observed in some natural phenomena. Similar to seismicity rate (see [31–33] for visual comparison) or volcano magma chamber inflation [34], BTC prices hold on to about 30–40% of the peak price after the pricing episode is over. The price decrease being of only approx. 1/3 of the asset volatility suggests that the confidence placed into the asset is never completely dissipated. The driving cause of price peak remains however unclear, and will require more research.

## 3.4. Similarities with assets under speculative pressure

On the financial market, rapid changes of asset prices may be explained by changes of confidence in the asset (e.g. bad and good results, scandal), by a temporary change in the liquidity of the asset, and by a variety of price manipulation schemes (e.g. pump and dump, insider trading, rumour propagation). In order to explore whether BTC price is driven by a fundamental cause or by external perception, we searched within financial market data if the constant time development of 50–70 days could be observed for other assets of various liquidity. Here, we state that an asset is liquid when any amount of the asset can be traded in a cash market without materially affecting its price. We also assume an orderly transaction in the sense of fair value measurement as defined by IFRS 13, i.e. a transaction is not forced and the agent making the transaction is able to conduct usual marketing activities (such as gathering a sufficient number of competitive bids). In that context, the liquidity can be interpreted as a measure of confidence, seeing how the seller–confident that the price of the same assets will be marginally changed in a near-future–is not afraid of losing wealth by selling their assets as they are. And of course, one should keep in mind that the level of investors' confidence may be impacted at any time by changing market context (central banks interest rates raises, stock market volatility, etc.).

Because of their known broad liquidity regimes and/or high demand and/or speculation histories, we selected analogues of BTC time series such as gold (XAU), currencies pairs (TRY,

**Table 1. Characteristics of time series and statistics of fit curves (Fig 5).** The Root Mean Square (RMS) difference between the modeled curve and original time-series never exceeds a ~26%. Average length of subdataset is approx. 190 days (median ~ 180).

| Event | T_1 | T_2 | P_min | P_max | Price change ($US) | Date 1 (dd mm yyyy) | Date 2 (dd mm yyyy) | RMS | RMS Red (%) | Panel in Fig 5 |
|---|---|---|---|---|---|---|---|---|---|---|
| 1 | 89 | 186 | 0 | 1 | 1 | 13 Nov. 2010 | 18 Feb. 2011 | 0.08 | 85.07 | a) |
| 2 | 199 | 261 | 1 | 4 | 3 | 03 Mar. 2011 | 04 May 2011 | 0.2 | 87.32 | b) |
| 3 | 199 | 299 | 1 | 35 | 34 | 03 Mar. 2011 | 11 June 2011 | 2. | 76.34 | c) |
| 4 | 517 | 699 | 4 | 13 | 8 | 15 Jan. 2012 | 15 Jul. 2012 | 0.9 | 86.02 | d) |
| 5 | 727 | 937 | 10 | 229 | 219 | 12 Aug. 2012 | 10 Mar. 2013 | 6 | 85.30 | e) |
| 6 | 1017 | 1174 | 66 | 1132 | 1065 | 29 May 2013 | 02 Nov. 2013 | 28 | 91.19 | f) |
| 7 | 1917 | 2099 | 365 | 705 | 339 | 15 Nov. 2015 | 15 May 2016 | 21 | 95.17 | g) |
| 8 | 2191 | 2467 | 596 | 2953 | 2357 | 15 Aug. 2016 | 18 May 2017 | 165 | 86.78 | h) |
| 9 | 2454 | 2524 | 1933 | 4066 | 2133 | 05 May 2017 | 14 Jul. 2017 | 421 | 84.60 | i) |
| 10 | 2191 | 2647 | 596 | 17803 | 17206 | 15 Aug. 2016 | 14 Nov. 2017 | 1033 | 74.51 | j) |
| 11 | 2999 | 3204 | 3236 | 11007 | 7770 | 01 Nov. 2018 | 25 May 2019 | 1044 | 80.83 | k) |
| 12 | 3561 | 3728 | 9048 | 19104 | 10055 | 16 May 2020 | 30 Oct. 2020 | 1312 | 89.10 | l) |
| 13 | 3561 | 3773 | 9048 | 40789 | 31740 | 16May 2020 | 14 Dec. 2020 | 2503 | 84.70 | m) |
| 14 | 3561 | 3815 | 9048 | 57533 | 48484 | 16 May 2020 | 25 Jan. 2021 | 5621 | 75.25 | n) |

and EUR), equities (ENRON, EXCITE), CC (ETHEREUM), bond yields (Greece 10-YR yields) and Tulip bulbs in the 17th century (Figs 6 and 7). Those assets of various classes span the complete range of liquidity in the market sense. Some are considered highly liquid (Gold-XAU), others experienced prominent periods of illiquidity (Turkish lira and Greek debt), others yet have reached terminal illiquidity (Enron and Excite). At last, we chose Ethereum as a reference because of its different governance mechanism and because of the high correlation between ETH and BTC prices ($r^2 = 0.91$, N = 180) over the last six months. Prices variations of those asset values have of course different causes (from purely speculative versus business plan revaluation) but their consequences are very similar: quick change of price followed by a stagnation, and then a sudden price readjustment after more information becomes available to a wider audience.

For the periods following each peak, one can distinguish two classes of assets: those which deflate completely (or return to their normal value as Greece 10-YR bond yield) and those which hold their value for various periods of time (XAU, TRY, BTC). In all cases, confidence plays a strong role in the price fixed by the market, and a given level of illiquidity is reached close to the price peak time.

Amongst this group of assets, we distinguish between those who experience illiquidity due to high demand (energy trader ENRON, search web engine EXCITE) and those with

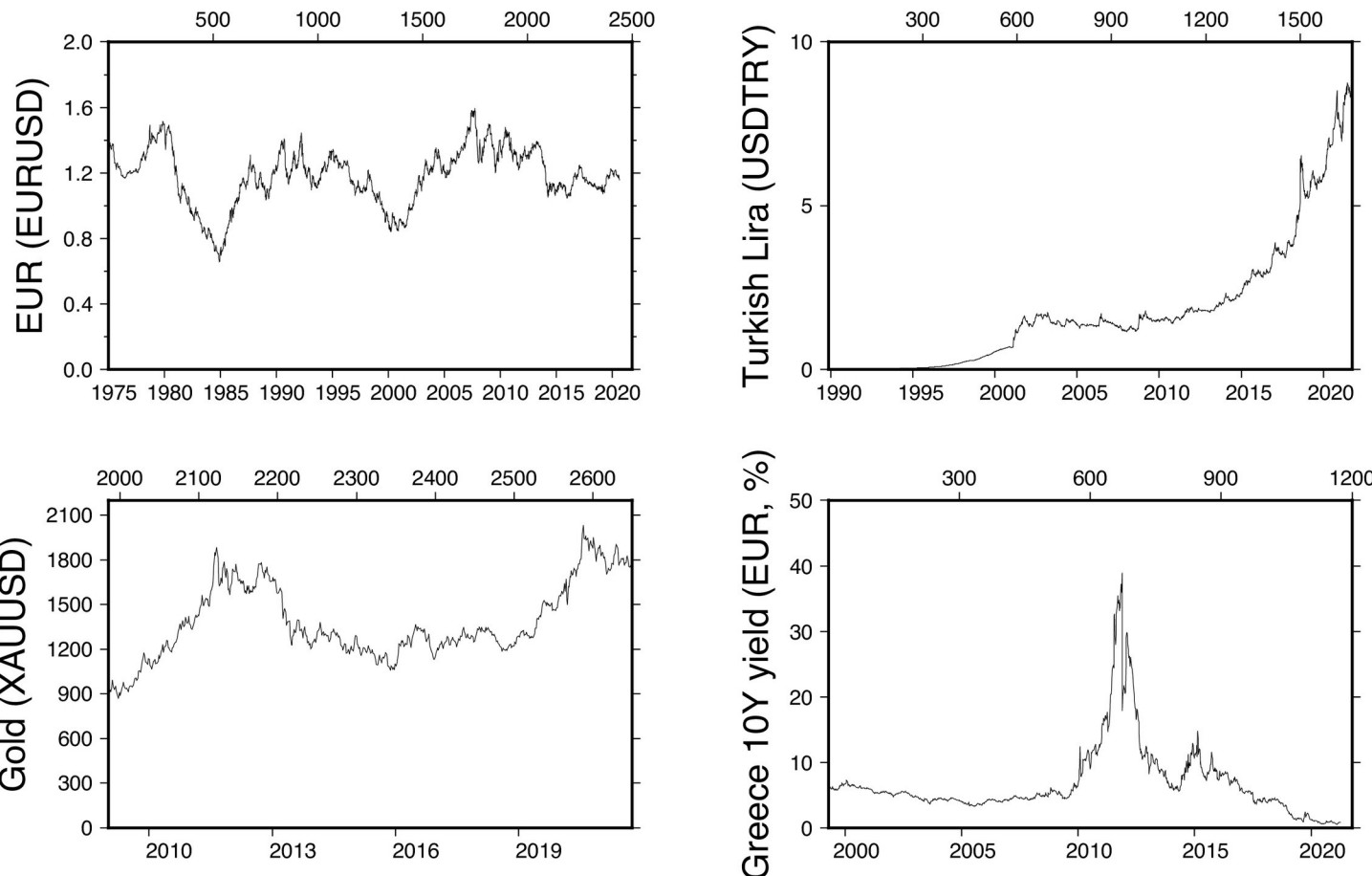

**Fig 6. Time series of Gold (XAU), currencies (TRY and EUR) and 10YR Greek bond yields (weekly data).** Currencies and bond yields show very diverse patterns due to different economic, market and political contexts. Only 10yr Greek Gov. bond yields time-series shows similarities with BTC pricing episodes until the peak is reached.

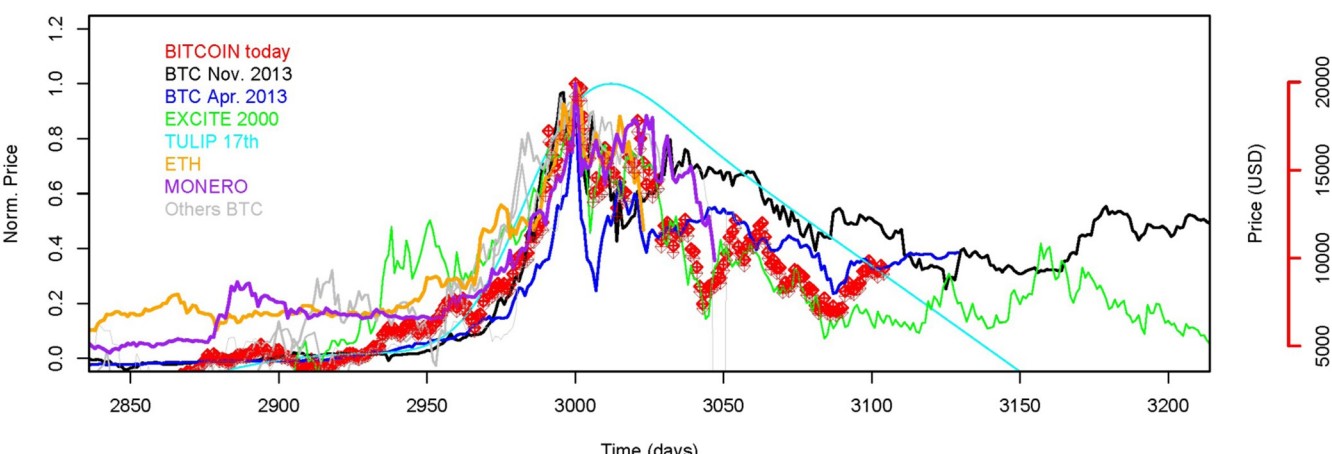

**Fig 7. Time-shifted (x-axis) and normalized (y-axis) time-series corresponding to data shown in Fig 1.** We compare BTC price to BTC pricing periods, Excite stock price, Tulip bulb crisis, and ETH/MONERO prices observed in 2017–2018.

extremely low demand (TRY, Greeks). Meanwhile, some did not reach their lowest level due to confidence loss (TRY, XAU, Greeks), and others did because of fundamental business issues (Excite) or even accounting fraud (Enron) and, in the end, were revealed as valueless. Our comparison shows that BTC is not fundamentally different from assets under speculative pressure. What is specific to BTC is that, as for gold (XAU), the stabilization of prices following a price peak suggests that investors' confidence level remains strong despite the intrinsic parameters indicate the risk of holding BTC for investor is high.

## 3.5. Correlation with other CC

For other types of assets, it has been observed that a sudden price rise for a given product may spill to others in the same sector; an effect amplified with the level of media attention [35–38]. Capital spilling suggests that investors aim at investing in alternative assets which are either cheaper or more liquid. This is also true for the most capitalized virtual currencies. News stories about the BTC performance increased the visibility of other crypto-currencies, and encouraged the risk-seeking investors to make a compromise between fund allocated and reputation of the CC in order to buy assets as cheap as possible while benefiting from the herd effect. If media attention and social network activity may impact the price of all CC, it seems that differences in design (i.e. centralization, number of coins available) might not have an influence on their price dynamics. Like in the past, contagion was made easier by the availability of common trading tools for a wide variety of trading financial products. The high correlation between virtual assets in general, and their correlated returns following a mediatic event (e.g., NBC Saturday Night Live) confirm that the level of confidence of investors plays a large role in the pricing of virtual assets.

## 4. Discussion

In this study we examine episodes of BTC price surges. We found that, in the past, BTC sudden price increases have lasted less than 100 days, were not followed by a full depreciation, BTC staying instead at a level close to 30–50% of its peak value, and that the successive BTC price changes were of quasi-exponential nature. We now hope that, when more data become

available, analyses similar to those already applied to other kinds of financial returns [39, 40] shall be carried out for BTC and other CC. Put simply, BTC has a lot in common with hard-to-borrow assets, mostly because the market liquidity is limited during periods of price inflation. The valuation of BTC, however, remains a complex endeavor, and one should expect it to behave like any other investment instrument under the scrutiny of a wide population of investors. The overall consistency of each BTC pricing event might be what makes BTC unusual compared to other financial pricing events; few stocks experienced consecutive crises of increasing amplitude without disappearing, or suffering so much that their values never recovered. Finally, we have shown that observations made on the BTC price history could be extended to other types of assets (crypto or not), despite fundamental differences in governance models and price structures (centralization, emissions strategies, price models, underlying businesses). As the BTC also behaved like other equities under speculative pressure, BTC should not be seen purely as a virtual currency. This observation is also supported by the numerous uses of BTC by various owners (savings, speculative, purchase of services, political aims, taste for technologies, hedging, diversification, long-term investment etc.) as published in the recent literature.

But BTC prices are not only driven by pulses of trading. BTC price dynamics is also sensitive to causes outside its own pricing mechanisms (mining, validation, number of coins on the market). Those causes are regulatory framework(s), media attention, social media activity, market conditions, and this open list may be expanded in the future. The relative contribution of all effects is likely sensitive to international market conditions and technological sector influence. All along the BTC price history, prices and traded volumes were very much related to the opening/closing of trading platforms with the implementation of national regulations. For instance, when traded volumes were the largest, during the fall 2017, new platforms based in China, with more relaxed rules regarding funds origins and traders identifications, were very active [15]. From January 2018, trading volumes were dramatically reduced, from millions of coins traded daily to thousands, suggesting that while the price stabilized in the range of $8'000–10'000, the price was not driven by the amount of BTC traded nor by the media coverage (see Google Trends time-series in Supplementary Materials). Finally, new regulations focusing on publicizing the identities of traders and owners, or the provenance of funds, with an aim to prevent illegal use of coins, could obviously change the dynamics of CC in the near-future, as suggested before [41].

Regarding media (social or not), trading volume increases observed in 2017–2018 were comparable to those observed in the 1980's in other contexts [42, 43]. This research described sudden price increases followed by warnings of market makers and regulators, resulting in the fall of the stock price of interest. Such a loss of enthusiasm in financial assets has been observed in the past, for instance during the bursting of the dot-com bubble, or following press releases on company performance. The fact that BTC survived various episodes of confidence loss during the last decade demonstrates that it is not purely speculative. Rather, its behavior results from a combination of owners' trust in the future of BTC [44, 45], safety of the transaction system (block chain), and public interest into the asset [11, 46–50].

Whilst our observations are supported by more than 10 events over a decade and more than five order to price magnitude, some questions remain. We are not able to predict the time and amplitude of the next price peak. Also, we found that in some cases it is difficult to discern the starts and ends of peaks when they are close to each other (Fig 5. m/n). A more sophisticated analysis may be helpful in finding the origin of those "split peaks", and also in linking trading volumes, platform activities and prices on platforms. Further research may help identify potential bottlenecks (trading delays, wrong prices, etc.) between banks involved in derivative products emissions and crypto-platforms trading coins which are used as hedge by those banks.

During our research, we faced some issues to explain our results from an economic perspective, because of the lack of research in some domains. First, further research should be carried around the role of platforms within the trading environment (banks, exchanges, retail investors, institutional investors), including in the Over-The-Counter (OTC) trades as initiated by [51]. As an extension, it would be useful to determine the floating quantity and the tracking of coins in order to constrain which portion of the asset is considered as reserve or long-term investment. In the financial domain, it would be necessary to establish clearly whether BTC prices (and CCs prices generally) correlate with other asset class prices, and over

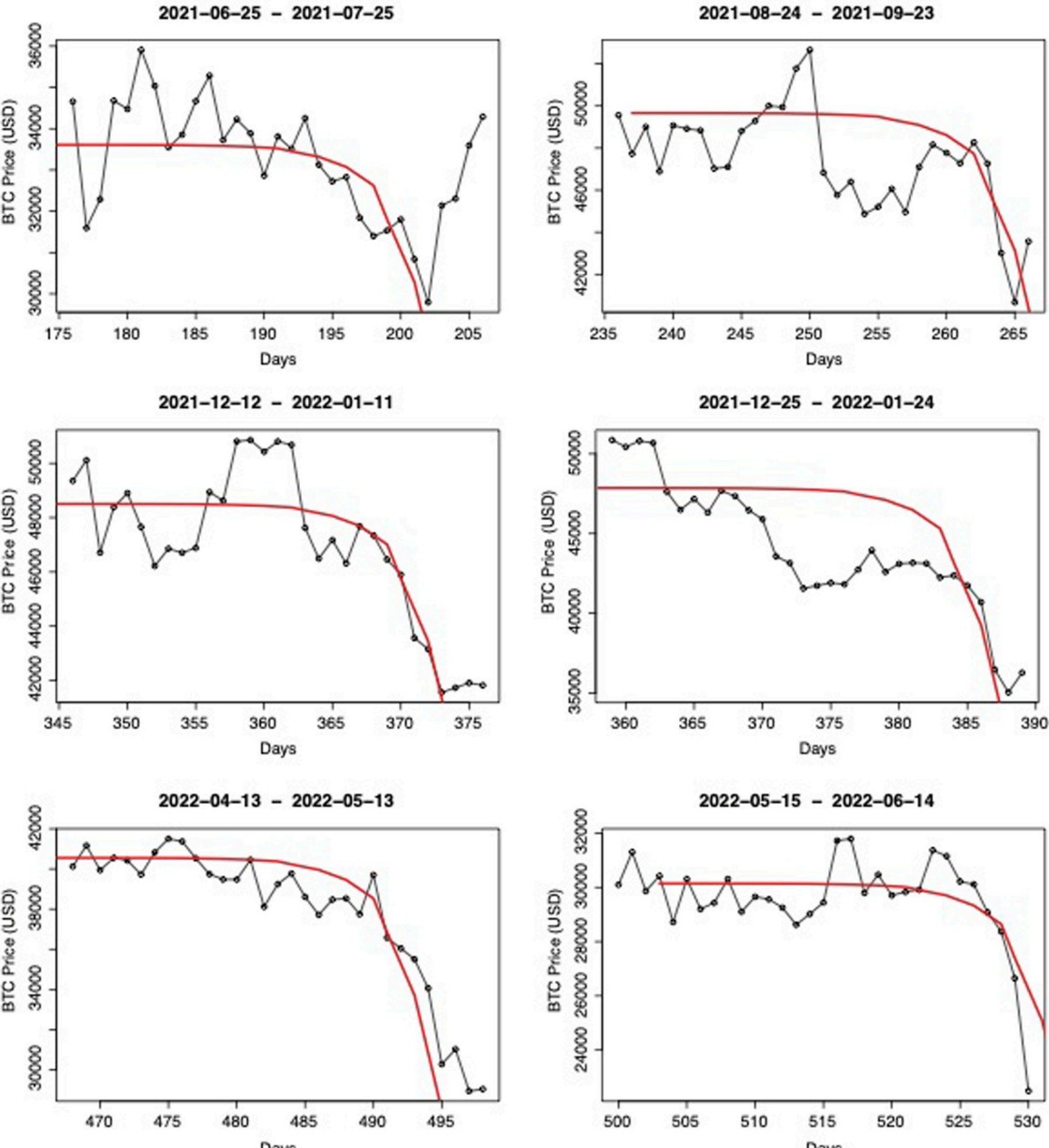

**Fig 8. Preliminary search for price decreases for the period 01 Jan. 2021–14 Jun. 2022.** We show that the hockey stick function could explain price decreases as well; although the time of development is shorter (days) and likely rooted in the intra-day trading activity.

what time-scale. Regarding exchanges efficiency, we could not explore intra-day price varia-tions because we were not able to access the necessary data so far. Those data are of particular importance to document price decrease episodes which span usually less than 5 days (see Fig 8 for episodes between 01 Jan. 2021 and 15 Jun. 2022). Finally, it would be highly useful to con-tinue studying the sociological profile of the crypto investor (e.g.: age, date of entry in the crypto market, wealth level, country, trade volumes) as such information may help banks define the risk appetite of investors, provide better services, and guaranty the stability of the trading environment [52].

## Supporting information

**S1 Fig. Periodogram for the sinus function;** $f = \sin(t/30)$**.**
(PDF)

**S2 Fig. Periodogram for the sinus function;** $f = \sin(t/30)$ **plus a step (**$dz = 0.1$**) at** $t > 1000$**.**
(PDF)

**S3 Fig. Periodogram for the sinus function;** $f = \sin(t/30)$ **plus a step (**$dz = 1.5$**) at** $t > 1000$**.**
(PDF)

**S4 Fig. Periodogram for the sinus function [**$f = \sin(t/30)$**] plus an additional sinus [**$f = \sin((t\text{-}60)/5)$**] at** $t > 1000$**.**
(PDF)

**S5 Fig. Periodogram for the equity stock Wirecard (WDI).**
(TIFF)

**S6 Fig. Periodogram for the equity stock ABB (ABB).**
(TIFF)

**S7 Fig. Periodogram for the equity stock Tesla (TSLA).**
(TIFF)

**S8 Fig. Periodogram for the equity stock GameStop (GME).**
(TIFF)

**S9 Fig. Periodogram (zoom) for the equity stock GameStop (GME).**
(PDF)

**S1 Dataset. Google trend map by city (search = 'bitcoin price').**
(CSV)

**S2 Dataset. Google trend map by country (search = 'bitcoin price').**
(CSV)

**S3 Dataset. Google trend time series (search = 'bitcoin price').**
(CSV)

**S4 Dataset. Google trends related searches to search = 'bitcoin price'.**
(CSV)

## Acknowledgments

N.H. thanks Pr. Dr. A. Berentsen (Uni. Basel), Dr. Christine Lang (FINMA) and 6 anonymous reviewers for their useful comments.

## Author Contributions

**Conceptualization:** Nicolas Houlié.

**Data curation:** Nicolas Houlié.

**Formal analysis:** Jevgeni Tarassov, Nicolas Houlié.

**Methodology:** Jevgeni Tarassov.

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
