## [Decision Letter · Decision Letter 0]

21 Jul 2021

PONE-D-21-18020

BITCOIN: a life in crisis

PLOS ONE

Dear Dr. Houlie,

Thank you for submitting your manuscript to PLOS ONE. After careful consideration, we feel that it has merit but does not fully meet PLOS ONE’s publication criteria as it currently stands. Therefore, we invite you to submit a revised version of the manuscript that addresses the points raised during the review process.

Though Reviewer 2 rejected PONE-D-21-18020, the reviewer provided many valuable and constructive comments. Considering three reviewers’ useful comments and the interesting topic of the manuscript, I would like to give you a chance to revise your manuscript during the special period. The revised manuscript will undergo the next round of review by the same reviewers. 

We look forward to receiving your revised manuscript.

Kind regards,

Baogui Xin, Ph.D.

Academic Editor

PLOS ONE

Journal Requirements:

4. Please upload a copy of Figure A3, to which you refer in your text on page 5 and 17. If the figure is no longer to be included as part of the submission please remove all reference to it within the text.

Reviewers' comments:

Reviewer's Responses to Questions

**Comments to the Author**

1. Is the manuscript technically sound, and do the data support the conclusions?

Reviewer #1: Partly

Reviewer #2: Partly

Reviewer #3: Yes

2. Has the statistical analysis been performed appropriately and rigorously? 

Reviewer #1: No

Reviewer #2: No

Reviewer #3: Yes

3. Have the authors made all data underlying the findings in their manuscript fully available?

Reviewer #1: Yes

Reviewer #2: Yes

Reviewer #3: Yes

4. Is the manuscript presented in an intelligible fashion and written in standard English?

Reviewer #1: Yes

Reviewer #2: Yes

Reviewer #3: Yes

5. Review Comments to the Author

Reviewer #1: This is an interesting paper, however, after reviewing the paper, I do not think it is appropriate for publishing. The paper still needs to be comprehensively improved before publishing. Some detailed comments are shown in the attachment file.

Reviewer #2: The questions raised in this article are interesting, but the methods used are not sufficient to support the conclusions reached, and the conclusions are not analyzed in depth. In addition, the writing of the paper is not standardized, the literature is not reviewed, and the marginal contribution of the paper is not highlighted.

Main concerns

1. Compared with previous articles on Bitcoin prices, what is the motivation and contribution of this article?

2. This paper found that the increase of Bitcoin price follows a similar “Hockey-Stick Shaped” pattern, but the authors did not analyze what caused this pattern, what characteristics of Bitcoin price this pattern reflects, and what implications the discovery of this pattern has on predicting Bitcoin prices.

3. This paper suggested that Bitcoin price is not only driven by the number of coins to be mined but also depends on both the investor’s confidence into the asset and the level of media attention. However, in the article, the authors did not provide data on investor confidence and media attention, but just made assumptions based on previous literature conclusions. This is not the attitude that academic research should have.

4. This paper recognized that Bitcoin should be classified as an illiquid asset, but this conclusion is inconsistent with the public perception. In particular, the authors’ argument for insufficient liquidity is insufficient, and it is difficult to persuade others to agree with the conclusion.

5. The authors of the paper should firmly grasp the purpose of the research, adopt sophisticated methods, reliable data, and fully demonstrate and analyze the conclusions.

Reviewer #3: The paper study the price change of BTC and found they follow a similar patten which called "Hockey-Stick Shaped" events. I find the paper very interesting . It can be accepted after some revision.

The data set is not detailed enough. The data set description section says that the data used in this paper is from 2017， but in the abstract the data set is related to 2018-2020.

6. PLOS authors have the option to publish the peer review history of their article (what does this mean?). If published, this will include your full peer review and any attached files.

Reviewer #1: No

Reviewer #2: No

Reviewer #3: No

---

## [Decision Letter · Decision Letter 1]

9 Dec 2021

PONE-D-21-18020R1BITCOIN: a life in crisesPLOS ONE

Dear Dr. Houlie,

Thank you for submitting your manuscript to PLOS ONE. After careful consideration, we feel that it has merit but does not fully meet PLOS ONE’s publication criteria as it currently stands. Therefore, we invite you to submit a revised version of the manuscript that addresses the points raised during the review process.

Though Reviewer 2 rejected PONE-D-21-26729, the reviewer provided many valuable and constructive comments. Considering three reviewers’ useful comments and the interesting topic of the manuscript, I would like to give you a chance to revise your manuscript during the special period. The revised manuscript will undergo the next round of review by the same reviewers.

We look forward to receiving your revised manuscript.

Kind regards,

Baogui Xin, Ph.D.

Academic Editor

PLOS ONE

Reviewers' comments:

Reviewer's Responses to Questions

**Comments to the Author**

1. If the authors have adequately addressed your comments raised in a previous round of review and you feel that this manuscript is now acceptable for publication, you may indicate that here to bypass the “Comments to the Author” section, enter your conflict of interest statement in the “Confidential to Editor” section, and submit your "Accept" recommendation.

Reviewer #1: (No Response)

Reviewer #2: All comments have been addressed

2. Is the manuscript technically sound, and do the data support the conclusions?

Reviewer #1: Yes

Reviewer #2: No

3. Has the statistical analysis been performed appropriately and rigorously? 

Reviewer #1: Yes

Reviewer #2: No

4. Have the authors made all data underlying the findings in their manuscript fully available?

Reviewer #1: Yes

Reviewer #2: Yes

5. Is the manuscript presented in an intelligible fashion and written in standard English?

Reviewer #1: Yes

Reviewer #2: Yes

6. Review Comments to the Author

Reviewer #1: This is an interesting research paper regarding the Bitcoin market. After reviewing the whole manuscript, I think the paper needs to do some necessary revisions before the official publishing.

1. Discussion should be the part to illustrate the rationality of the results. However, in the current manuscript, the current discussion seems to the outlook for the future. I suggest the authors to revise the current discussion.

2. Descriptions should make some revisions. For example, “Finally” in Line 277 and “Finally” in Line 280.

Reviewer #2: This article investigated the Bitcoin price dynamics and found that there were eight events with similar durations during the sample period. Studying the price of Bitcoin is a very interesting topic, but I think the authors’ research is not innovative enough and contributions are not enough, and the research design is not rigorous enough.

Main concerns

1. The authors should emphasize the research motivation. Why are you writing this article? What conclusion do you want to get?

2. The conclusion of the article is not credible. The author said that a function similar to the Fibonacci sequence can be used to approximate the price of Bitcoin. I find it hard to believe, and the author did not make further analysis.

3. The research design of the article is not rigorous. The authors’ conclusions all come from observations and intuitions of price dynamics, and they have not analyzed and discussed the conclusions in an economic sense. In addition, the author did not do a robustness test. Will the conclusion of the paper still hold for the period after January 1, 2020?

4. The writing of the paper is also not standardized. The table in the text is not a standard three-line table, the letters in the formula are not explained further, and the abscissa of the time series graph is not indicated by date. Therefore, I think this is not a qualified academic paper.

7. PLOS authors have the option to publish the peer review history of their article (what does this mean?). If published, this will include your full peer review and any attached files.

Reviewer #1: No

Reviewer #2: No

---

## [Editor Report · Decision Letter 2]

27 Jan 2022

PONE-D-21-18020R2

BITCOIN: a life in crises

PLOS ONE

Dear Dr. Houlie,

Thank you for submitting your manuscript to PLOS ONE. After careful consideration, we have decided that your manuscript does not meet our criteria for publication and must therefore be rejected.

Although you have an interesting and valuable paper, the paper needs to be substantially improved before it can be considered for PLOS ONE. Since one of reviewers still advise us to reject your manuscript, I will save your time and hope you can consider improving the manuscript accordingly and resubmitting it as a new article. And I would like to be the Academic Editor of the revised version again.

I am sorry that we cannot be more positive on this occasion, but hope that you appreciate the reasons for this decision.

Yours sincerely,

Baogui Xin, Ph.D.

Academic Editor

PLOS ONE
---

## [Author Response · Author response to Decision Letter 2]

1 Apr 2022

We have answered to latest comments. We have also attached the materials linked to the appeal.

---

## [Decision Letter · Decision Letter 3]

2 Jun 2022

PONE-D-21-18020R3BITCOIN: a life in crisesPLOS ONE

Dear Dr. Houlie,

Thank you for submitting your manuscript to PLOS ONE. After careful consideration, we feel that it has merit but does not fully meet PLOS ONE’s publication criteria as it currently stands. Therefore, we invite you to submit a revised version of the manuscript that addresses the points raised during the review process.

Though Reviewer 5 rejected PONE-D-21-18020R3, the reviewer provided some valuable and constructive comments. Considering two reviewers' useful comments and the interesting topic of the manuscript, I would like to give you a chance to revise your manuscript during the special period. The revised manuscript will undergo the next round of review by the same reviewers.

We look forward to receiving your revised manuscript.

Kind regards,

Baogui Xin, Ph.D.

Academic Editor

PLOS ONE

Reviewers' comments:

Reviewer's Responses to Questions

**Comments to the Author**

1. If the authors have adequately addressed your comments raised in a previous round of review and you feel that this manuscript is now acceptable for publication, you may indicate that here to bypass the “Comments to the Author” section, enter your conflict of interest statement in the “Confidential to Editor” section, and submit your "Accept" recommendation.

Reviewer #4: All comments have been addressed

Reviewer #5: All comments have been addressed

2. Is the manuscript technically sound, and do the data support the conclusions?

Reviewer #4: Yes

Reviewer #5: Partly

3. Has the statistical analysis been performed appropriately and rigorously? 

Reviewer #4: Yes

Reviewer #5: No

4. Have the authors made all data underlying the findings in their manuscript fully available?

Reviewer #4: Yes

Reviewer #5: Yes

5. Is the manuscript presented in an intelligible fashion and written in standard English?

Reviewer #4: Yes

Reviewer #5: Yes

6. Review Comments to the Author

Reviewer #4: This is an interesting paper. It is well motivated and generally well written. The topic is interesting and the results can have interesting policy implications for both policymakers and investors. Compared with the previous two versions, I can see the quality of this paper improves a lot. I therefore recommend that the paper be published in PLOS ONE. Congratulations to the authors.

Reviewer #5: It seems a very simple use of some equations. The topic at the beginning of the papar is very interesting but fail to be technically sound. I do not find a contribution around some diffeent ideas or matehmatically different. The conclusions are very naive.

7. PLOS authors have the option to publish the peer review history of their article (what does this mean?). If published, this will include your full peer review and any attached files.

Reviewer #4: No

Reviewer #5: No

---

## [Author Response · Author response to Decision Letter 3]

21 Jun 2022

please see attached files.

Reviewer #4: All comments have been addressed

Reviewer #5: All comments have been addressed

ANSWER 01: From this comment we consider we do not need to answer the comments attached to the current revision request. It seems to us that this file is inherited from another revision set and these comments have been answered in September 2021. We consider this as a mistake of the editor assistant. However, we update our answer to question 5 as highlighted in the cover letter.

Reviewer #4: This is an interesting paper. It is well motivated and generally well written. The topic is interesting and the results can have interesting policy implications for both policymakers and investors. Compared with the previous two versions, I can see the quality of this paper improves a lot. I therefore recommend that the paper be published in PLOS ONE. Congratulations to the authors.

ANSWER 02: We thank R1 for his positive feedback. We continued to research in the domain of price reduction since the last version. We show this latest finding in the new version and mention this in the discussion of the paper. 

Reviewer #5: It seems a very simple use of some equations. The topic at the beginning of the papar is very interesting but fail to be technically sound. I do not find a contribution around some diffeent ideas or matehmatically different. The conclusions are very naive.

ANSWER 03: We proposed in our paper that price increases and stabilization were driven at least partially by confidence. Recent events (price fall) have shown that confidence is driving the dynamics of the BTC price (wrt. Luna) and BTC price was not free being impacted by adverse market conditions (central banks interest rises). Since the last version of the manuscript, we have added a figure showing that our model is also true for price decreases (including the one of 15 Jun 2022). We thank R5 for his comments and hope the latest additional improvements will allow decrease the perceived level of naiveness of our research.

---

## [Decision Letter · Decision Letter 4]

24 Aug 2022

BITCOIN: a life in crises

PONE-D-21-18020R4

Dear Dr. Houlie,

I've been trying to support non-economists to employ some non-econometric approaches from their scientific fields to explore financial problem from some different perspectives. I also believe that my efforts mentioned above will better promote the diversified development of economics. I can understand Reviewer 6's motivation and worry because Reviewer 6 fears that this paper falls into the same trap that many similar mainstream currency papers trigger. To sum up, to promote the diversified development of economics, I still recommend this manuscript to be accepted.

We’re pleased to inform you that your manuscript has been judged scientifically suitable for publication and will be formally accepted for publication once it meets all outstanding technical requirements.

Kind regards,

Baogui Xin, Ph.D.

Academic Editor

PLOS ONE

Additional Editor Comments (optional):

Reviewers' comments:

Reviewer's Responses to Questions

**Comments to the Author**

1. If the authors have adequately addressed your comments raised in a previous round of review and you feel that this manuscript is now acceptable for publication, you may indicate that here to bypass the “Comments to the Author” section, enter your conflict of interest statement in the “Confidential to Editor” section, and submit your "Accept" recommendation.

Reviewer #5: All comments have been addressed

Reviewer #6: (No Response)

Reviewer #7: All comments have been addressed

2. Is the manuscript technically sound, and do the data support the conclusions?

Reviewer #5: Yes

Reviewer #6: Partly

Reviewer #7: Yes

3. Has the statistical analysis been performed appropriately and rigorously? 

Reviewer #5: Yes

Reviewer #6: No

Reviewer #7: Yes

4. Have the authors made all data underlying the findings in their manuscript fully available?

Reviewer #5: Yes

Reviewer #6: Yes

Reviewer #7: Yes

5. Is the manuscript presented in an intelligible fashion and written in standard English?

Reviewer #5: Yes

Reviewer #6: Yes

Reviewer #7: Yes

6. Review Comments to the Author

Reviewer #5: Comments and onservations have been included in the document. As a future line of research maybe you can develop this research for ore cryptocurrencies

Reviewer #6: I fear that this paper falls into the same trap that many similar mainstream currency papers trigger: applying time series methodologies and callibration techniques to financial price data to find a pattern, without a detailed analysis of the components and dynamics of the underlying systems contributing to the price behaviour. The universe of time series functions is sufficiently large that something can usually be found, but this doesn´t really help with understanding the behaviour of the underlying system. This is a particularly nasty trap in BTC´s case, since very limited data is available about other components of its price behaviour, in particular the amount of explicit and disguised fractional reserve banking that is being performed with BTC (f.ex. tether). Without a clear justification based on possible shared mechanisms, which is absent, I don´t think it is valid to blindly apply techniques from seismological or meterological data in the way suggested here. (I would invite the authors to consider their opinion of a paper which took methods from financial analysis and calibrated them to fit short term seismic data.)

As an aside, since the Nyquist theorm applies to the frequency analysis here, this implies that at least twice the time period of the underlying system/data must be available in order to avoid aliasing. The relevant time period for Bitcoin, if there is one, is not known. If the economic "business cycle" period, which is typically estimated between 10-20 years is taken as a rough subsitute, then BTC hasn´t been in existence long enough for any conclusion to be drawn from that form of analysis.

Overall there are a number of occasions in the paper where I think claims are being made without sufficient substantation, and that the paper is also not robust to a suggestion of some degree of cherry picking. For example, why Greek bonds in particular? There is a very large set of different bond instruments of different ratings to be selected from here, has an analysis been done on all of them? Similarly Enron is an interesting example of an instrument of systemic fraud, but not the only one from that period. No evidence is provided to support the assertation that Chinese mining was responsible for the peak in bitcoin volumes in 2016-17, and other causes have been suggested for this, including Tether manipulation, which is also more likely given the size of the deviation. Probably the most likely explanation for the price behaviour after price peaks is the sale of coins derived from mining to finance electricity and other related costs, but this factor is not mentioned.

A general problem with any financial time series analysis (which often gets overlooked) is finding satisfactory methods to compensate for changes in the underlying quantities of the unit of measurement (money). In principle, this is something that bitcoin was supposedly designed to avoid, but in practice this has not been the case so far. To this point in time (mid-2022) the supply of bitcoin has expanded relatively rapidly (as designed), and also nothing about bitcoin´s design prevented it from being used as an asset for fractional reserve lending, and as recent events have shown, this feature of the existing financial system duely emerged within the cryptocurrency financial system and has proceeded to wreak its own brand of financial chaos. Since the changes in quantity of BTC due to mining are now (as designed) stabilising, this should at least be highlighted in the paper as previous behaviour may not be a guide to future dynamics now that this is finally occurring.

Correcting (normalising) for changes in the quantity of the unit of measurements involved here (USA M3 and BTC) would make some of these affects clearer, and probably make some of the BTC price behaviour reported here more extreme, but would still run into the problem mentioned by the authors in the discussion in that there is also a very large amount of dormant BTC to be factored in. Whilst I think there might be an interesting data paper here, in terms of presenting the various time series and quantitative analysis that the author have performed, and an interesting historical paper, if a timeline of critical bitcoin events was added, for example the bitcoin mining rate date changes, first posting on slackdot.com, etc., I don´t find the financial price time series analysis compelling, or adding anything to the literature.

Reviewer #7: The paper investigates a hot topic in finance nowadays.

Reading the reviewers’ comments and the authors’ answers and the latest version of the paper, I found it interesting and it shed light to some aspects in the history of Bitcoin prices that worth investigation.

The authors mention they investigate the “BTC price time-series (17 August 2010 – 27 June 2021)” and “All data listed in this article can be found here: https://finance.yahoo.com/”

However, nowadays, in Yahoo finance data is not available for the entire period – older than 2014. In Yahoo finance “Date shouldn't be prior to '2014-09-17'”

The affirmation on row 275 (page 13) – “as published in the recent literature” – should be followed by some citation(s).

7. PLOS authors have the option to publish the peer review history of their article (what does this mean?). If published, this will include your full peer review and any attached files.

Reviewer #5: No

Reviewer #6: **Yes: **Jacky Mallett

Reviewer #7: No

---

## [Editor Report · Acceptance letter]

20 Sep 2022

PONE-D-21-18020R4 

BITCOIN: a life in crises 

Dear Dr. Houlié:

I'm pleased to inform you that your manuscript has been deemed suitable for publication in PLOS ONE. Congratulations! Your manuscript is now with our production department. 

Kind regards, 

on behalf of

Professor Baogui Xin 

Academic Editor

PLOS ONE